# A Convenient and Highly Efficient Strategy for Esterification of Poly (γ-Glutamic Acid) with Alkyl Halides at Room Temperature

**DOI:** 10.3390/polym17010010

**Published:** 2024-12-25

**Authors:** Youhong Ai, Yangyang Zhan, Dongbo Cai, Shouwen Chen

**Affiliations:** 1Hubei Collaborative Innovation Center for Advanced Organic Chemical Materials & Key Laboratory for the Synthesis and Application of Organic Functional Molecules (Ministry of Education), College of Chemistry & Chemical Engineering, Hubei University, Wuhan 430062, China; 2State Key Laboratory of Biocatalysis and Enzyme Engineering, Environmental Microbial Technology Center of Hubei Province, College of Life Sciences, Hubei University, Wuhan 430062, China; yangyangzhan@hubu.edu.cn (Y.Z.); caidongbo@hubu.edu.cn (D.C.)

**Keywords:** poly (γ-glutamic acid), esterification, alkyl halides, 1,1,3,3-tetramethylguanidine, poly (γ-glutamic acid) alkyl ester

## Abstract

The presented work discusses the highly efficient esterification of poly (γ-glutamic acid) (γ-PGA) with alkyl halides at room temperature. The esterification reaction was completed within 3 h, and the prepared γ-PGA esters were obtained with excellent yields (98.6%) when 1,1,3,3-tetramethylguanidine (TMG) was used as a promoter. The influence of the amount of TMG, solvent, reaction conditions, and alkyl halides on the esterification reaction was examined. It was found that polar aprotic solvents, such as *N*-Methylpyrrolidone (NMP) and 1,3-Dimethyl-2-imidazolidinone (DMI), were favorable for the esterification. Non-polar or weakly polar solvents (i.e., dichloroethane, acetonitrile) were not favorable for the esterification. Water as a solvent had a negative effect on esterification. The reactivity of bromine halogenated compounds was higher than that of chlorine halogenated compounds but lower than that of iodine halogenated compounds. The structures of the prepared γ-PGA ester were confirmed by ^1^H NMR and FT-IR spectroscopy. Thermal stability and hydrophobic properties of the resulting product were tested. The results showed that the prepared γ-PGA propyl ester had high thermal stability (up to 267 °C) and showed good hydrophobicity (contact angle 118.7°).

## 1. Introduction

Poly (γ-glutamic acid) (γ-PGA) is a biopolymer consisting of L- and/or D-glutamic acid monomers polymerized through γ-glutamyl bonds [1]. Due to its versatile properties of hygroscopicity, water solubility, biodegradability, and non-toxicity towards humans and the environment, γ-PGA has been widely used in various fields, including foods, medicine, cosmetics, and agriculture industries [2]. γ-PGA is mainly produced by chemical synthesis, extraction from natto, and microbial fermentation. Compared with chemical synthesis and extraction, microbial fermentation is simple and suitable for large-scale production. The strains producing γ-PGA are mainly concentrated in Bacillus species such as Bacillus subtilis, Bacillus licheniformis, and Bacillus amyloliquefaciens [3,4,5]. Although γ-PGA had a good application prospect, its application was greatly limited by its low thermal stability, poor processing properties, and strong hydrophilicity. Esterification modification for γ-PGA would be a better way to comprehensively solve the above problems. γ-PGA is a polycarboxylic acid whose carboxylic groups are able to undergo a chemical modification. γ-PGA can be esterified with halogenated hydrocarbons and alkyl diazo compounds, and the esterified compounds are hydrophobic [6]. γ-PGA can react with alkyl dihalides, diamines, epoxy resins, and other crosslinking agents to form γ-PGA crosslinking gels [7]. γ-PGA can also be blended with a variety of polymers to achieve complementary properties of polymers, such as improving the hydrophilicity and biocompatibility of the polymer, so as to realize the purpose of modification.

The carboxyl groups of γ-PGA can be converted into hydrophobic alkoxy carboxyl groups or benzyl carboxyl groups by esterification modification [8], and its solubility in organic solution can be improved. By looking up the literature, there were two main esterification methods for γ-PGA: direct esterification and transesterification. For direct esterification, the reaction conditions were not very harsh. First, under the reaction condition of 60 °C, γ-PGA was dissolved in dimethyl sulfoxide (DMSO), and then excess NaHCO_3_ was added to neutralize γ-PGA into carboxylate, which was subsequently desalted with excess haloalkane or benzyl bromide for 19 h; finally, products were precipitated with cold hydrochloric acid solution, washed with anhydrous ethanol, and dried under vacuum to obtain white γ-PGA ester in powder form. Kubota et al. [9,10,11] carried out the hydrogen halogenated elimination reaction between γ-PGA and alkyl halide to esterify part of the carboxyl group and obtained the corresponding alkyl-γ-PGA ester. The esterified groups could be aliphatic alkyl groups, alicyclic alkyl groups, aryl groups, etc. The resulting γ-PGA ester was easy to form into a membrane and had excellent strength, transparency, and strong elasticity. The esterification degree of this direct esterification method was low, and the concentration of NaHCO_3_ also had an effect on esterification during the reaction. Too high a concentration could degrade γ-PGA, and the esterification degree decreased with the extension of the alkyl side chain. In addition, the reaction time of this esterification took generally 5–7 days. In order to overcome the shortcomings of direct esterification, Morillo et al. [12,13] proposed a transesterification method (two-step synthesis of γ-PGA ester). The main process was as follows: first, γ-PGA was esterified into γ-PGA ethyl ester under the action of bromoethane, and then the γ-PGA ethyl ester was dissolved in the alkanols, and corresponding γ-PGA esters were obtained by transesterification reaction at 180 °C and nitrogen with tetrabutyl titanate as catalyst. The esterification degree of the product obtained by this method was higher than that by direct esterification, and the product was easier to purify. However, both direct esterification and transesterification often required a higher reaction temperature and longer reaction time, which led to higher costs. Sometimes it was difficult to obtain a satisfactory reaction yield. Therefore, it is highly desired to develop a convenient and highly efficient approach for the esterification of γ-PGA under mild conditions in a short time.

1,1,3,3-Tetramethylguanidine (TMG) is a colorless liquid with excellent solubility. As a strong organic base catalyst, TMG has been widely used in industry, mainly as a co-solvent for the synthesis of new antibiotic cephalosporins and also as a widely used catalyst in the synthesis of polyurethanes [14]. At the same time, tetramethylguanidine has been used in organic synthesis [15,16,17,18] and gene transfection [19,20]. In 1989, Kocienski et al. [21] first reported that TMG used in the synthesis of zoapatanol promoted the esterification of carboxyl compounds with iodomethane in benzene and found that 77% of the yield was obtained after the reaction at 20 °C for 2 h. Subsequently, this reaction was also used by Tanaka [22] and Kocienski [23] for a carboxyl methylation reaction in organic synthesis, but the yield was only maintained at about 70%. Li et al. developed a facile and highly efficient approach for the esterification of benzoic acid with halogenated compounds promoted by TMG, and the method was extended to the esterification of polyacrylic acid and polymethylacrylic acid [24,25,26]. Poly(meth)acrylates with a 100% degree of esterification were obtained in DMSO at room temperature.

Considering the versatile properties of TMG in organic reactions, herein, we attempted the esterification of γ-PGA promoted by TMG under mild conditions. The aim of this work was to develop a convenient and highly efficient method for the esterification of γ-PGA to reduce the costs of esterification and obtain esterification products with high thermal stability and hydrophobicity. The illustration of the esterification of γ-PGA is presented in Figure 1. We expect that the resulting γ-PGA esters could have good applications in food, agriculture, and the pharmaceutical industry.

## 2. Materials and Methods

### 2.1. Materials and Measurements

γ-PGA with an average molecular weight of 1.02 × 10^6^ Da and polar aprotic solvents N-Methylpyrrolidone (NMP), *N*,*N*-Dimethylformamide (DMF), DMSO, and 1,3-Dimethyl-2-imidazolidinone (DMI) were purchased from the Aladdin Chemistry Co. (Shanghai, China). Alkyl halides, 1,1,3,3-tetramethylguanidine, sodium bicarbonate, and absolute ethyl alcohol were purchased from Sinopharm Chemical Reagent Co., Ltd. (Shanghai, China). All other chemicals were of analytical or reagent grade and used directly without further purification.

Fourier-transform infrared (FT-IR) absorption spectra were recorded with a Nexus-670 Fourier-transform infrared spectrometer (Nicolet, Massachusetts, USA). ^1^H-NMR spectra were measured with a 400 MHz nuclear magnetic resonance instrument (Ascend TM 400, Bruker, Bremen, Germany). The thermogravimetric analysis (TGA) was carried out on a SETSYS-16 (TGA Setaram, Décines-Charpieu, France), and the temperature was changed from room temperature to 600 °C at a heating rate of 10 °C/min in a nitrogen atmosphere. The contact angle (CA) of water on the coating surface was measured with a Kruss DSA-100 interface tensiometer (Hamburg, Germany).

### 2.2. Preparation of Polyglutamate Sodium

The pH of the aqueous solution of γ-PGA was adjusted to 3 by hydrochloric acid, and the aqueous solution was precipitated with 4 times the volume of 95% ethanol. The sediment was dissolved with distilled water and then precipitated again with 95% ethanol, and the precipitate was collected, which was γ-PGA (pH about 3). After γ-PGA was dissolved again, solution was neutralized with NaOH (pH = 7). The product was poly (γ-glutamic acid) sodium (γ-PGA-Na).

### 2.3. Procedure of Esterification Reaction

Taking the esterification of γ-PGA with propyl bromide in NMP solvent as an example, the procedure of the esterification reaction was as follows: The previously prepared γ-PGA-Na (768 mg, 6 mmol) was suspended in 50 mL of NMP. The mixture was stirred at 60 °C for 30 min to give a homogeneous solution. Then the mixture was cooled to room temperature (about 30 °C). Propyl bromide (3.69 g, 30 mmol) was dissolved in the mixture solution, followed by the addition of TMG (1.15 g, 10 mmol), and vigorous stirring was continued at room temperature for 3 h. The reaction was terminated by placing the reaction solution in a refrigerator, and the precipitated sodium bromide was removed by decantation. The supernatant was poured into 300 mL of absolute ethyl alcohol under stirring. The precipitate was collected by centrifugation and washed with a large amount of absolute ethyl alcohol several times. The isolated polymer was dried in vacuo at 40 °C for 12 h to obtain γ-PGA propyl ester. Because the esterification product is insoluble in anhydrous ethanol, and ethanol can be well miscible with the reaction solvent (NMP, DMI, etc.), the product in the reaction solvent can be precipitated by the addition of a large amount of ethanol. The structures of the product were confirmed by ^1^H NMR and FT-IR spectroscopy. The yield was determined by the gravimetric method. The procedure for the esterification reaction is as shown in Figure 1. For comparison, the esterification of γ-PGA with propyl bromide in NMP solvent was performed in the absence of TMG under the same conditions as above.

### 2.4. Measurement of the Yield of Esterification

Because esterification occurred on the carboxyl groups of the repeating structural units of γ-PGA, the yield was determined by the gravimetric method. The esterification of γ-PGA with propyl bromide was taken as an example; the yield could be calculated by the formula mentioned below, where 151 is the molecular weight of the repeating structural unit of γ-PGA-Na, 171 is the molecular weight of the repeating structural unit of γ-PGA propyl ester, *m*_0_ is the initial mass of γ-PGA-Na in the reaction solution (mg), and *m* is the mass of the product γ-PGA propyl ester (mg). Here, all carboxyl groups in γ-PGA were thought to be involved in esterification under the conditions of this experiment.
(1)$$Yield=\frac{151\times m}{171\times m_{0}}\times 100\%$$

## 3. Results and Discussion

### 3.1. Characterization of the Product of Esterification

NMP was chosen as a solvent for the esterification reaction. The esterification reaction of γ-PGA with propyl bromide was conducted at room temperature by using TMG as a promoter for 3 h. After careful purification, the resulting product was obtained as a clear, colorless solid. The product was characterized by its FT-IR spectrum in KBr and its ^1^H NMR spectrum in DMSO-d_6_. As shown in Figure 2, the peak at 1730.44 cm^−1^ can be attributed to the typically strong absorption peak of the ester carbonyl group (R-C(=O)-O-R’), the peak at 2962.30 cm^−1^ can be assigned to the stretching vibration from methyl and methylene groups in propyl groups in the polymer, the peak at 1647.59 cm^−1^ corresponds to the -C=O stretching vibration from carbonyl groups in the main chain of the polymer, the peak at 3088.16 cm^−1^ is attributed to the N-H stretching vibration, and the peak at 2933.28 cm^−1^ could be assigned to the C-H stretching vibration of γ-PGA-Na and γ-PGA propyl ester. These are consistent with those of the γ-PGA propyl ester as reported in the literature [11].

In the ^1^H NMR spectrum shown in Figure 3, the signal at 4.3 ppm is characteristic and ascribed to the ester-linked methylene protons (OCH_2_); the signals at 1.68 and 1.06 ppm are assigned to methylene (-CH_2_-) and methyl (-CH_3_) protons in propyl, respectively, of the γ-PGA propyl ester unit. Since the relationship of their respective integral values is 2:2:3, these are assigned to the protons of the propyl groups marked e, f, and g. The other protons in the molecule also have corresponding ascription. ^1^H-NMR (DMSO-d_6_, 400 MHz): δ 1.06 (t, 3H, CH_3_), 1.68 (m, 2H, CH_2_), 1.92 (m, 2H, CH_2_), 2.19 (t, 2H, α-CH_2_), 3.52 (m, 1H, CH), 4.30 (t, 2H, OCH_2_), and 8.19 (d, 1H, NH). These results clearly indicate that γ-PGA was successfully esterified with propyl bromide in NMP at room temperature by using TMG as a promoter. In addition, in the ^1^H NMR spectrum, the relative area of the proton signal peak is proportional to the number of equivalent hydrogen atoms that produce the signal peak. The intensity ratio of ^1^H NMR signals of propyl protons in the side chain and methylene protons in the γ-PGA ester main chain is consistent with the repeating structural unit of γ-PGA propyl ester. Therefore, all carboxyl groups in γ-PGA were esterified under the conditions of this experiment (the degree of esterification of γ-PGA was 100%). The signal for an unknown impurity in the ^1^H NMR spectrum is negligibly small.

### 3.2. Esterification Kinetics

The esterification of γ-PGA-Na with propyl bromide promoted by TMG as a model reaction and the relationship between reaction time and reaction progress were studied. According to the calculation formula of the reaction yield proposed in the experimental section, the reaction yields under different reaction times were obtained, and the reaction kinetics curve as shown in Figure 4 was plotted. From the reaction kinetics curve, in the absence of TMG, the yield was less than 10% within 3 h and was only 67.9% when the reaction time was extended to 24 h. However, when using TMG as a promoter, the reaction yield reached more than 60% after 1 h, and the entire esterification reaction could be completed within 3 h. We could intuitively find that the esterification reaction at room temperature promoted by TMG was more efficient than that in the absence of TMG. Moreover, compared with the esterification method mentioned in the literature [10,11], which required higher reaction temperature and longer reaction time, the proposed esterification promoted by TMG was more convenient and efficient. The esterification reaction could be successfully completed in a relatively short time with a high yield. A possible reason was proposed. Under the promotion of TMG, γ-PGA-Na was easy to form carboxylate negative ions, and then carboxylate negative ions attacked the α-carbons in alkyl halides to form ester groups, and at the same time, the halogen atom transferred to form sodium salt, which would be conducive to the rapid esterification reaction.

In addition, the concentration of TMG can also affect the progress of esterification. The amount of TMG added was changed from 0 g to 1.5 g to evaluate its effect, and the results are shown in Appendix A. As can be seen, the yield of the esterification increased with the addition of TMG. The yield reached a maximum when the amount of TMG was 1.15 g, and then it remained almost unchanged. So, the amount of TMG was set at 1.15 g in the following experiments.

### 3.3. Effect of Solvents on Esterification Reaction

It is well known that solvents often have a great effect on a reaction [27]. The influence of solvents on the esterification reaction promoted by TMG was studied. In general, the esterification of carboxyl groups of polymers with alkyl halides follows the bimolecular substitution reaction mechanism (Sn2). A strong polar aprotic solvent is favorable for the esterification [24]. Several polar aprotic solvents, including NMP, DMF, DMSO, and DMI, were used. In addition, non-polar or weakly polar solvents (i.e., dichloroethane, acetonitrile, and purified water) were compared as supporting solvents for the esterification of γ-PGA-Na. It was found that the γ-PGA-Na was not dissolved well in solvents (i.e., DMF, DMSO, dichloroethane, and acetonitrile) at 30 °C, so the esterification reaction was carried out at 60 °C. The effects of solvents on the reaction were assessed according to the yield of the obtained product. The results are summarized in Table 1. As a result, the esterification reaction carried out in NMP and DMI solvents gave better yield. When using DMSO, the obtained yield was low. The probable reason might be the presence of the Kornblum oxidation reaction in the esterification reaction solution [28]. DMSO could be used as an oxidizing agent to oxidize alkane halides to corresponding carbonyl compounds, which was not conducive to the esterification reaction. When DMF was present as a reaction solvent, the yield was also low due to the poor solubility of γ-PGA-Na in the DMF solvent. Although γ-PGA-Na had good solubility in purified water, the esterification product was not obtained even after 19 h. This was because protic solvent (H_2_O) had strong solvation with the carboxylate anion and suppressed the reactivity of the anion as a nucleophile, while in polar aprotic solvents (NMP or DMI), the anion was not solvated and was “naked”; therefore, the reaction efficiency was high [29]. Moreover, the esterification product was insoluble in water, which also had an adverse effect on the esterification reaction. This was the same in polar solvents like methanol and ethanol. It is evident from Table 1 that polar aprotic solvents such as NMP and DMI are particularly favorable for this esterification reaction, while in non-polar and weakly polar solvents, the reaction efficiency becomes worse.

### 3.4. Effect of Halide Structure on Esterification Reaction

We studied the esterification of γ-PGA-Na with different halides (including chloride, bromide, and iodide) in NMP solvent at room temperature promoted by TMG. After 2.5 h of reaction at room temperature, the esterification yield of ethyl iodide was 97%. After 3 h of reaction, the yield of 1-bromopropane was 98%. However, after 5 h of reaction, the yields of 1-chloropropane and benzyl chloride were 73% and 62%, respectively. These showed that the esterification reactivity of these halides with similar structure was different, and the reactivity was as follows: iodide > bromide > chloride.

The effect of alkyl bromide homologues on esterification reactions was also studied. Several *n*-alkyl bromide homologues were selected for esterification with γ-PGA-Na. The reactions were conducted in NMP at room temperature using TMG as a promoter, and the results are shown in Appendix A. Esterification reactions of *n*-alkyl bromide homologues could proceed to obtain the corresponding esters with excellent yield of esterification. When *n*-butyl bromide was used, the yield of esterification was 95.5% in 3.5 h, and *n*-decyl bromide only achieved an 89.5% yield of esterification in 4.5 h, which indicated that the esterification efficiency slowed down gradually with the increase in the R group in alkyl bromides. This was mainly due to the reason that the R group in alkyl bromides inhibited the bimolecular attack of the carboxylate ions in γ-PGA-Na on the alkyl bromide from the back through its own steric hindrance effect. This is consistent with the fact that the TMG-promoted esterification is a bimolecular nucleophilic substitution reaction (S_N_2).

### 3.5. Thermal Stability and Hydrophobic Properties of γ-PGA Propyl Ester

Previously, Kubota reported that the free acid type of γ-PGA melted at 210 °C, and thermal degradation of the γ-glutamyl chain concurrently started [11]. The improvement in the thermal stability of γ-PGA ester by the proposed esterification was expected. Thermal properties of γ-PGA propyl ester and γ-PGA-Na were compared by TGA as shown in Figure 5. Decomposition of γ-PGA-Na started at about 221 °C. γ-PGA propyl ester melted and decomposed at 267 °C. The high thermal stability of the esterification product was considered to be beneficial to the molding and processing performance of the products. So, esterification promoted by TMG is a simple and effective strategy to obtain esterification products with high thermal stability.

Hydrophobicity of γ-PGA is another critical factor determining its applications as packaging material for food. Due to the large number of hydrophilic carboxyl groups in γ-PGA molecules, its water solubility leads to certain limitations in the application of this material. γ-PGA was usually compounded with other materials to improve its performance or was modified to overcome its water solubility [30]. The improvement in the hydrophobicity of the esterification product was expected also. Here, it was evaluated through the contact angle (CA) of water on the obtained esterification product. As shown in Appendix A, the CA of water on the γ-PGA-Na surface is 63.2°, while the CA of water on the γ-PGA propyl ester surface is 118.7°, which indicates that γ-PGA propyl ester is hydrophobic and suitable for application as packaging material for food. This hydrophobicity can be ascribed to the introduction of alkyl groups into γ-PGA molecules by this esterification, which makes the surface Gibbs free energy of γ-PGA-Na decrease and reduces the contact between water and γ-PGA-Na. It could be predicted that the hydrophobicity of the γ-PGA alkyl ester will gradually increase with the increase in the introduced alkane group chain by this esterification.

### 3.6. Evaluation of Esterification Methods

The esterification method of γ-PGA was evaluated, and the results are listed in Table 2. Compared with other esterification methods reported, the proposed method provides a comparably higher yield, less reaction time, and a lower reaction temperature.

Moreover, when we expanded the quantity of reactants (a few grams to tens of grams), we found that the esterification reaction could still smoothly proceed and the yield could still reach more than 98%. There were two main reasons for these. The esterification of γ-PGA promoted by TMG was very efficient, and the purification of the esterification product was easy. Therefore, this mild and efficient esterification method will have a good application prospect.

## 4. Conclusions

We used TMG as a promoter for the esterification of γ-PGA with alkyl halides in polar aprotic solvents, such as NMP and DMI. The experimental results demonstrated that the esterification reaction was very rapid and efficient with an excellent yield of esterification at room temperature. The reaction behaviors of different halogenated compounds were quite different. The reactivity of bromine halogenated compounds was higher than that of chlorine halogenated compounds but lower than that of iodine halogenated compounds. Esterification reactions of *n*-alkyl bromide homologues could proceed to obtain the corresponding esters with excellent yield of esterification. The esterification efficiency slightly slowed down with the increase in the R group in alkyl bromides. In addition, the obtained esterification product showed high thermal stability and hydrophobicity. Therefore, the esterification of γ-PGA promoted by TMG will create new opportunities for the γ-PGA carboxylic esters to have good economic value and application prospects.

## Figures and Tables

**Figure 1 polymers-17-00010-f001:**
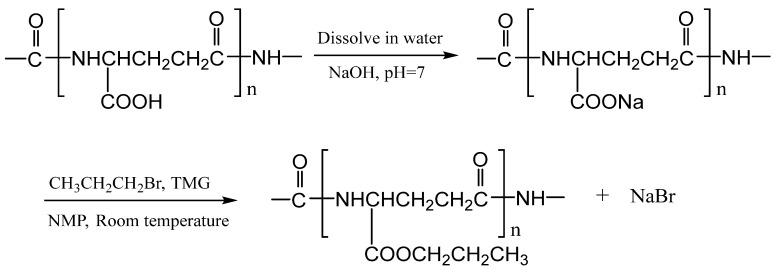
The illustration of the esterification of poly (γ-glutamic acid) with propyl bromide at room temperature promoted by 1,1,3,3-tetramethylguanidine (TMG).

**Figure 2 polymers-17-00010-f002:**
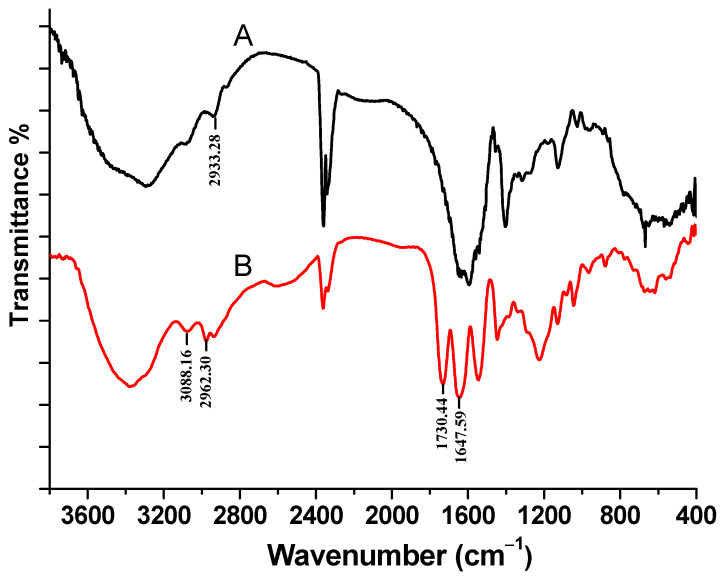
FT-IR spectrum of hydrophobic γ-PGA-Na (A) and γ-PGA propyl ester (B) obtained by esterification of γ-PGA with propyl bromide using TMG as a promoter in NMP at room temperature.

**Figure 3 polymers-17-00010-f003:**
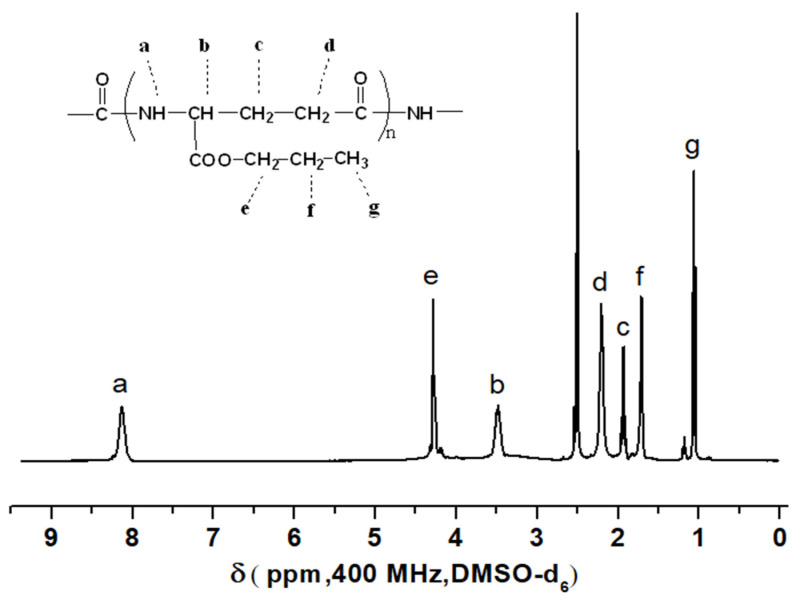
^1^H NMR spectrum of γ-PGA propyl ester obtained by esterification using TMG as a promoter in NMP at room temperature (DMSO-d_6_, 400 MHz).

**Figure 4 polymers-17-00010-f004:**
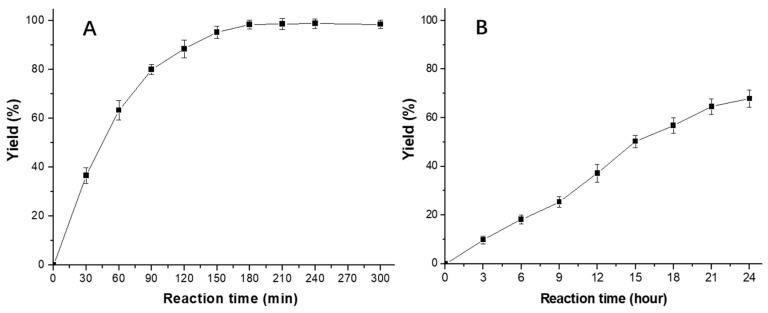
Plots of product yields versus reaction time for the esterification reaction of γ-PGA-Na (768 mg) with propyl bromide (3.69 g) in NMP (50 mL) at room temperature using TMG (1.15 g) as a promoter (**A**) and in the absence of TMG (**B**).

**Figure 5 polymers-17-00010-f005:**
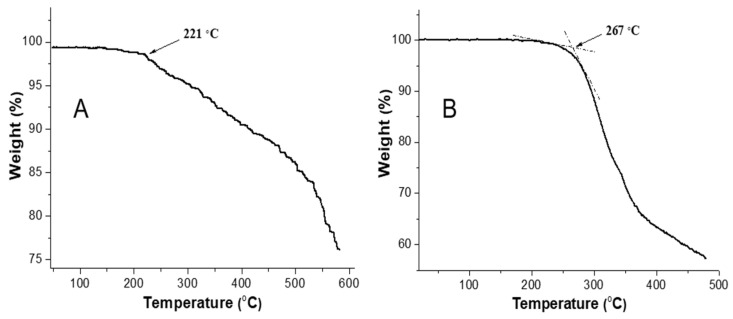
Thermogravimetric analysis of γ-PGA-Na (**A**) and γ-PGA propyl ester (**B**). TGA was measured at a heating rate of 10 °C/min.

**Table 1 polymers-17-00010-t001:** Esterification of γ-PGA-Na with propyl bromide in various solvents using TMG as a promoter ^a^.

Entry	Solvents	Temperature/°C	Time/h	Yield/%
1	NMP	30	3	98.6 ± 0.72 ^b^
2	DMF	60	19	61.3 ± 1.20
3	DMSO	60	19	68.7 ± 0.71
4	DMI	30	3	92.8 ± 1.32
5	Dichloroethane	60	19	23.2 ± 0.97
6	Acetonitrile	60	19	18.5 ± 0.54
7	Water	30	19	No precipitate

^a^ Reaction conditions: γ-PGA-Na (768 mg), propyl bromide (3.69 g), solvent (50 mL), and TMG (1.15 g) as a promoter. The yield of esterification was determined by the gravimetric method. ^b^ Relative standard deviation.

**Table 2 polymers-17-00010-t002:** Comparison of several esterification methods of γ-PGA.

Esterification Method	Promoted	Solvent	Reaction Temperature	Reaction Time	Degree of Esterification	Yield	Ref.
Direct esterification	TMG	NMP	Room temperature	3 h	100%	98.6%	This work
Direct esterification	/	NMP	60 °C	19 h	55%	73%	[9]
Direct esterification	/	NMP	60 °C	19 h	100%	77.9%	[10]
Transesterification	/	DMSO	180 °C	10 h	98%	90%	[11]

## Data Availability

The datasets generated for this study are available on request to the corresponding author.

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
