# Peer review of "A Convenient and Highly Efficient Strategy for Esterification of Poly (γ-Glutamic Acid) with Alkyl Halides at Room Temperature"

_polymers, 2024, doi:10.3390/polym17010010_

Round 1

Reviewer 1 Report

Comments and Suggestions for Authors

1.      The abstract needs to highlight all the key observations of the work, Need to be re-written

2.      Reference should be provided for converting carboxyl groups of γ-PGA into alkoxy or benzyl carboxyl groups by direct esterification. Page 2

3.      TMG has excellent solubility in which type of solvent?

4.      Full form of the abbreviation should be used at first instant

5.      For FTIR the Y-axis should be %Transmittance

6.      For FTIR author should mention all the important peaks for both A and B

7.      What could be the chemistry behind the great reactivity when TMG is used?

Author Response

Dear Reviewer #1,

Thank you very much for your comments concerning our paper entitled:" A Convenient and Highly Efficient Strategy for Esterification of Poly (g-glutamic acid) with Alkyl Halides at Room Temperature "(Manuscript ID: polymers-3363553) and please transfer our appreciation to the Assistant Editor. According to your comments, we have revised the manuscript carefully. Uploaded is the revised manuscript (with changes marked), we hope it now can meet the requirements of polymers. As to the questions raised by the reviewer #1, we respond as follows.

Sincerely yours,

Youhong Ai

College of Chemistry & Chemical Engineering

Hubei University

Wuhan, 430062

P R China

e-mail: aiyouhong83@sina.com

Response to Reviewer #1:

  1. The abstract needs to highlight all the key observations of the work, Need to be re-written

Answer: Thank you very much for your comments and suggestions. All key observations from this work have been added to the abstract. All changes for the manuscript were marked in red font.

  1. Reference should be provided for converting carboxyl groups of γ-PGA into alkoxy or benzyl carboxyl groups by direct esterification. Page 2

Answer: According to your suggestion, we have provided the reference in the second paragraph of the introduction (see reference 8).

  1. TMG has excellent solubility in which type of solvent?

Answer: TMG is easily soluble in polar solvents, such as N-Methylpyrrolidone (NMP), N, N-Dimethylformamide (DMF), Dimethyl sulfoxide (DMSO) and 1,3-Dimethyl-2-imidazolidinone (DMI), etc.

  1. Full form of the abbreviation should be used at first instant

Answer: According to your suggestion, full form of the abbreviation was used at first instant.

  1. For FTIR the Y-axis should be %Transmittance

Answer: According to your suggestion, “Intensity” was replaced by “Transmittance” for FTIR.

  1. For FTIR author should mention all the important peaks for both A and B

Answer: According to your suggestion, several characteristic peaks have been marked in the FTIR, and the peaks have been assigned in the revised manuscript.

  1. What could be the chemistry behind the great reactivity when TMG is used?

Answer: We think that TMG maybe act as a catalyst. TMG participated in the esterification reaction, which reduced the activationenergy of the reaction and increased the reaction rate significantly.

Reviewer 2 Report

Comments and Suggestions for Authors

The authors in this paper have shown a convenient and highly efficient strategy for the esterification of poly(γ-glutamic acid) with alkyl halides using 1,1,3,3-tetramethylguanidine (TMG) as a promoter at room temperature. Overall, the manuscript is well-organized and written, with the results presented in an excellent way. This work will attract attention from researchers in the scientific community and the broad audience of Polymers.

However, I have a few minor questions that the authors can address in the revised manuscript:

  • Esterification yields were high (92 to 98.6%) in polar aprotic solvents and low in non-polar or weakly polar solvents. What are the side products from non-polar or weakly polar solvents, and how was the esterification product isolated?
  • Did the reaction result in a precipitate in water due to the protic solvent? Is this the same in polar solvents like methanol and ethanol?

Author Response

Dear Reviewer #2,

Thank you very much for your comments concerning our paper entitled:" A Convenient and Highly Efficient Strategy for Esterification of Poly (g-glutamic acid) with Alkyl Halides at Room Temperature "(Manuscript ID: polymers-3363553) and please transfer our appreciation to the Assistant Editor. According to your comments, we have revised the manuscript carefully. Uploaded is the revised manuscript (with changes marked), we hope it now can meet the requirements of polymers. As to the questions raised by the reviewer #2, we respond as follows.

Sincerely yours,

Youhong Ai

College of Chemistry & Chemical Engineering

Hubei University

Wuhan, 430062

P R China

e-mail: aiyouhong83@sina.com

Response to Reviewer #2:

Comments and Suggestions for Authors

The authors in this paper have shown a convenient and highly efficient strategy for the esterification of poly(γ-glutamic acid) with alkyl halides using 1,1,3,3-tetramethylguanidine (TMG) as a promoter at room temperature. Overall, the manuscript is well-organized and written, with the results presented in an excellent way. This work will attract attention from researchers in the scientific community and the broad audience of Polymers.

However, I have a few minor questions that the authors can address in the revised manuscript:

Esterification yields were high (92 to 98.6%) in polar aprotic solvents and low in non-polar or weakly polar solvents. What are the side products from non-polar or weakly polar solvents, and how was the esterification product isolated?

Answer: Thank you very much for your comments and suggestions. As γ-PGA-Na has a single maximum absorption peak at 200nm, UV spectrophotometry can be used to detect the content of unreacted γ-PGA-Na, so as to illustrate the progress of the reaction. The side products from non-polar or weakly polar solvents were not found in the experiment. Because the esterification product is insoluble in anhydrous ethanol, and ethanol can be well miscible with the reaction solvent (NMP, DMI etc.), the esterification products in the reaction solvent can be precipitated by the addition of a large amount of ethanol.

Did the reaction result in a precipitate in water due to the protic solvent? Is this the same in polar solvents like methanol and ethanol?

Answer: Water had adverse effect on the esterification reaction. This was because protic solvent (H2O) had strong solvation with the carboxylate anion and suppressed the reactivity of the anion as nucleophile. This was the same in polar solvents like methanol and ethanol.

Reviewer 3 Report

Comments and Suggestions for Authors

This work discusses the efficient esterification of poly (g-glutamic acid) (g-PGA) with alkyl halides at mild temperature. Solvent, temperature, alkyl halide was varied in order understand the Esterification reaction. Overall, the authors have done a careful work, and the presentation of results is clear. The introduction was nicely written and clearly sated the advantages of TMG used in the synthesis process. The conclusion was well written. There are however quite a few points that require the author's attention before the paper can be accepted for publication.

How do you calculate % substitution of Alkyl halide to the poly (g-glutamic acid) (g-PGA)? It would be highly informative if the author included a representative 1H NMR spectrum and provided details of the integration values used for the calculation.

Author Response

Dear Reviewer #3,

Thank you very much for your comments concerning our paper entitled:" A Convenient and Highly Efficient Strategy for Esterification of Poly (g-glutamic acid) with Alkyl Halides at Room Temperature "(Manuscript ID: polymers-3363553) and please transfer our appreciation to the Assistant Editor. According to your comments, we have revised the manuscript carefully. Uploaded is the revised manuscript (with changes marked), we hope it now can meet the requirements of polymers. As to the questions raised by the reviewer #3, we respond as follows.

Sincerely yours,

Youhong Ai

College of Chemistry & Chemical Engineering

Hubei University

Wuhan, 430062

P R China

e-mail: aiyouhong83@sina.com

Response to Reviewer #3:

Comments and Suggestions for Authors

This work discusses the efficient esterification of poly (g-glutamic acid) (g-PGA) with alkyl halides at mild temperature. Solvent, temperature, alkyl halide was varied in order understand the Esterification reaction. Overall, the authors have done a careful work, and the presentation of results is clear. The introduction was nicely written and clearly sated the advantages of TMG used in the synthesis process. The conclusion was well written. There are however quite a few points that require the author's attention before the paper can be accepted for publication.

How do you calculate % substitution of Alkyl halide to the poly (g-glutamic acid) (g-PGA)? It would be highly informative if the author included a representative 1H NMR spectrum and provided details of the integration values used for the calculation.

Answer: Thank you very much for your comments and suggestions. Here all carboxyl groups in γ-PGA were thought to be involved in esterification under the conditions of this experiment. The yield formula given in this paper was used to reflect the degree of esterification. Moreover, the intensity ratio of 1H NMR signal of propyl protons in side chain and methylene protons in γ-PGA ester main chain was consistent with the repeating structural unit of γ-PGA propyl ester. Therefore, all carboxyl groups in γ-PGA were esterified under the conditions of this experiment.